# Formal pyridine *meta*-azidation and its application for the synthesis of diazepines, ring-fused δ-carbolines and 1,2,3-triazolylpyridines

Shu-Min Guo [1,2], Pengwei Xu [1,2], Constantin G. Daniliuc [1] & Armido Studer [1] ✉

The nitrogen atom serves as an important structural element in pharmacologic studies, highlighting the critical role of nitrogen-containing heterocycles in drug discovery. Late-stage peripheral functionalization and structural editing of nitrogen-containing heterocycles have garnered increasing attention due to their potential for the preparation of diversifying drug-like libraries. In particular, the structural diversification of pyridine cores by introducing N atoms or N-containing functionalities shows promise, but remains underexplored. Here, we report a synthetic strategy that combines the introduction of the versatile azide moiety with molecular editing. This approach comprises an initial peripheral regioselective meta-azidation of pyridines through dearomatized oxazino pyridine intermediates, and a subsequent molecular editing step via a photo-mediated singlet nitrene insertion process. We have demonstrated the utility of this strategy by the synthesis of seven-membered diazepines. Furthermore, ring-fused δ-carbolines and 1,2,3-triazolylpyridines can be also accessed through chemical manipulation of the azide functionality.

Nitrogenated heterocycles are among the most prominent core scaffolds in pharmaceuticals (Fig. 1A).[1–3] In particular, pyridine has emerged as the most frequently occurring structural motif among FDA-approved drugs over the past decade.[4] By comparison, the exploration of other high-value compounds containing uncommon aza-ring systems—such as diazepines[5] and carbolines[6]—has been limited due to challenges in their synthesis and restricted accessibility. Over the past decades, organic azides have been considered as powerful reactive entities for the preparation of N-heterocyclic ring systems from basic building blocks.[7] Owing to their 1,3-dipole character and unique property to readily release nitrogen upon photolysis or pyrolysis, organic azides possess diverse chemical reactivities and appear as versatile synthetic precursors or N atom transfer reagents in chemistry and materials science (Fig. 1B).[8,9] For example, copper(I)-catalyzed azide-alkyne Huisgen 1,3-dipolar cycloaddition has been well-documented and accordingly widely applied in the synthesis of 1,2,3-triazole-based bioactive molecules and functional polymers (click chemistry).[10–12] In addition, organic azides are often employed as amino or nitrene equivalents in other important transformations, such as the Staudinger reaction, the Curtius rearrangement, and transition-metal catalyzed nitrene chemistry.[7]

Molecular editing of (hetero)arenes has recently come into focus as a powerful strategy to chemically modify the core structure of molecules at a late stage through peripheral functionalization or skeletal interconversion.[13–27] Nitrene mediated skeletal editing of (hetero)arenes were pioneered by Huisgen[28] and Doering[29], demonstrating benzene ring expansion via N-atom insertion under pyrolysis or photolysis conditions.[20,30] This strategy has since been successfully applied

---

[1]Organisch-Chemisches Institut, Universität Münster, Münster, Germany. [2]These authors contributed equally: Shu-Min Guo, Pengwei Xu.
✉e-mail: studer@uni-muenster.de

**Fig. 1 | Background and motivation. A** Diverse N-containing core structures in drugs. **B** Significance of the azide functionality in synthesis. **C** Reaction design. **D** This work: The sequential conversion of pyridines.

to various other (hetero)arenes, such as benzenes,[31–36] indoles[37] and pyrroles[38] by using different nitrene precursors. Unsurprisingly, the valuable strategy of molecular editing has also been applied to inherently electron-deficient pyridines, often focusing on their conversion to benzenes.[39–47] Recently, Zheng,[48] Ghiazza, and Moreau[49] independently reported the nitrogen insertion into pyridines to access 1,2-diazepines starting from pyridinium ylides.

Considering the versatility of organic azides, along with our background in molecular editing of azines,[22,50,51] we herein propose a sequential conversion of pyridines comprising an initial dearomatization with subsequent peripheral C−H functionalization to install the essential azide group at the meta-position that is followed by a photo-mediated molecular editing via a singlet nitrene insertion process (Fig. 1C). Notably, the inherent electron-deficient nature of pyridines makes the incorporation of nitrogen-containing functional groups highly challenging, in particular at the meta-position.[52] Thus, only a few examples for the meta-C−H nitration or amination of pyridines have been reported to date,[53,54] and the meta-azidation of pyridines remains unexplored.[55] In general, the synthesis of aryl azides requires a pre-installed functionality (for example, halogen, diazonium, amine group) as a leaving group or transient substituent.[8] These facts have hindered the exploration of meta-azido pyridine reactivity in the past. Temporary dearomatization has shown high potential for the synthesis of meta-substituted heteroarenes by transferring electron-deficient pyridines into electron-rich enamine type intermediates.[55–58]

In this work, we suggest the meta-C−H azidation of pyridines through dearomatized oxazino pyridine intermediates (Fig. 1D).[52,56] In the subsequent molecular editing step, the pyridyl azides can be further converted into the corresponding reactive nitrene intermediates upon irradiation. As a result, various interesting pharmacophores, including 1,3-diazepines, 1,4-diazepines, and δ-carbolines, become available under mild conditions starting with highly abundant pyridines. In addition, the meta-azido pyridines can also be transferred to the corresponding value-added 1,2,3-triazoles and meta-amino

pyridines. The proposed strategy should also be applicable to diversify the pyridine core in more complex drug compounds.

## Results
### Method development

The easily accessible sodium 2,2,6,6-tetramethylpiperidin-1-olate (TEMPONa) was applied by us as a single-electron-transfer (SET) reagent for the azidooxygenation of alkenes using the Zhdankin reagent **A** as the azidyl radical precursor under mild condition.[59] Motivated by these findings, we assumed that the electrophilic N_3 radical generated through SET-reduction of **A** by TEMPONa can be trapped by the nucleophilic dienamine entity of a dearomatized oxazino pyridine. Studies commenced with the 2-phenyl pyridine-derived oxazino pyridine **S1** (mixture of diastereoisomers) with Zhdankin reagent **A** and freshly prepared TEMPONa in THF at room temperature. We were pleased to obtain the azido-TEMPO-intermediate **Int-1** with 23% NMR yield as a mixture of diastereoisomers with complete regioselectivity (Table 1, entry 1). Reaction optimization showed that an excess of **A** and careful temperature adjustment are crucial to improve efficiency (entry 2–6). Screening of solvents revealed *t*BuOMe as the best solvent to afford **Int-1** with 96% isolated yield in a 0.4 mmol scale (entry 7–10). Importantly, the final rearomatizing hydrolysis step can be realized as a one-pot process, as demonstrated for meta-azidation of **S1** to afford **1** in overall 64% NMR yield (entry 11). Unfortunately, product loss was noted during the final rearomatization step by HCl-treatment. We assume that the hydrolysis step involves two competitive pathways (Fig. 2). In pathway **a**, TEMPOH is first fragmented to give azidated oxazino pyridine **Int-2** that is followed by acid-mediated aromatization to afford the targeted azide **1**. Alternatively, an iminium intermediate **Int-3** is generated first through pathway **b**, and TEMPOH elimination eventually leads to **1**. Due to significant product loss, we further investigated the hydrolysis step and converted **Int-1** into **Int-2** by treatment with the Fukukawa dehydrating reagent (see SI, Fig. S2).[60] **Int-2** was found to be highly unstable

**Table 1 | Reaction optimization for the meta-azidation of the phenyl-substituted oxazino pyridine S1**

| Entry | Variation from above | Conv. (%) | Yield of Int-1 or 1 (%)[a] |
|---|---|---|---|
| 1 | 1.5 equiv. **A**, THF, rt. | 57 | 23 (**Int-1**) |
| 2 | 2.0 equiv. **A**, THF, 0 °C | 37 | 30 (**Int-1**) |
| 3 | 2.0 equiv. **A**, THF, 30 °C | 44 | 35 (**Int-1**) |
| 4 | 2.0 equiv. **A**, THF | 55 | 44 (**Int-1**) |
| 5 | 2.0 equiv. **A**, THF, 50 °C | 63 | 45 (**Int-1**) |
| 6 | THF | 78 | 65 (**Int-1**) |
| 7 | PhMe | 87 | 65 (**Int-1**) |
| 8 | DCE | 100 | 56 (**Int-1**) |
| 9 | 1,4-dioxane | 91 | 47 (**Int-1**) |
| 10[b] | tBuOMe | 100 | 96[c] (**Int-1**) |
| 11 | tBuOMe/tBuOMe[e] | 100 | 64[d] (**1**) |
| 12 | tBuOMe/MeCN[e] | 100 | 77[d] (69[d,f]) (**1**) |

Reactions were performed on a 0.1 mmol scale. *TEMPO* 2,2,6,6-Tetramethylpiperidinyloxyl, *THF* Tetrahydrofuran, *DCE* 1,2-Dichloroethene.
[a]The yield was determined by ¹H NMR with dichloromethane as internal standard.
[b]Reaction was performed on a 0.4 mmol scale.
[c]Isolated yield of **Int-1**.
[d]Yield of 5-azido-2-phenylpyridine (**1**) obtained after hydrolysis; conditions: 1 mL 6 N HCl, 0.5 mL organic solvent, at 60 °C for 12 h.
[e]Solvent used for hydrolysis.
[f]Isolated yield.

and immediately reacted under hydrolysis conditions to azido pyridine **1**, albeit in 13% yield only. We currently believe that pathway **a** is the low-yielding transformation and should be suppressed. The solvent for the hydrolysis step was therefore carefully examined and we noted significant impact on the yield (see SI, Table S1). Among the solvents tested, MeCN afforded the highest yield (77% overall yield, entry 12).

**Formal peripheral meta-azidation of pyridines through their oxazino pyridines—reaction scope**

Under the optimized conditions, a broad range of oxazino pyridines were investigated and converted successfully to the corresponding meta-azido pyridines in moderate to good yields (Fig. 3). All oxazino pyridines used in this study were prepared according to literature[50,52,53,56,61] in 88% average yield (see SI, GP1). For ortho-substituted substrates (**S1**–**S5**), the regioselectivity of the azidation (C3 versus C5, nomenclature based on the starting pyridine) relies on the electronic properties of the ortho-aryl substituent, and in general, good yields were obtained (50–68%). The parent phenyl (**1**) and electron-poorer aryl substituted oxazino pyridines reacted with excellent regioselectivity at the C5 position (**2**–**4**, **6**), while **S5** carrying an electron-rich bisalkoxyaryl substituent provided after hydrolysis the C3 and C5 azido pyridine in equal amounts (**5**). The high C5 selectivity observed in most cases aligns with previously reported selectivities for the reaction of electrophilic radicals with oxazino pyridines.[56,61]

Substrates carrying functionalized phenyl groups (**7**–**9**), pyridyl (**10**), fused-ring systems (**11**–**12**), and alkyl groups (**13**–**16**) at the para-position afforded exclusive mono-azidation products with moderate to good yields (30–61%). Oxazino pyridine **S9** exhibited a lower reactivity (30% recovery), likely due to poorer nucleophilicity or increased steric hindrance from the nitro and methyl substituents. Notable, the benzothiophene moiety—being inherently more reactive than the pyridine counterpart—remained stable and compatible with our reaction conditions (**12**).

Considering that pyridine-derived drugs and bioactive molecules usually contain more than one substituent, 2,3-, 2,4-, and 2,5-disubstituted pyridines were evaluated and all delivered through the corresponding oxazino pyridines the targeted products (**17**–**28**, 22–51%). For oxazino pyridine **S19**, we attributed the lower reactivity (40% recovery of starting material) to the decreased nucleophilicity of the trifluoromethyl-substituted oxazino pyridine. For the methoxy derivative **18**, the rearomatization step was challenging, explaining the lowered yield. For the 2,4-disubstituted oxazino pyridine, azidation occurred with excellent regioselectivity at the C5 position (**22**).

To further probe the applicability of this method, late-stage transformation of more complex structures and drugs was performed. As examples, the glutamate receptor modulator VU6001966 (**29**), and vismodegib (**30**) could be meta-functionalized through the corresponding oxazino pyridines. In the latter case, both regioisomers were formed. Furthermore, meta-acylated nikethamide, more complex loratadine and tropicamide were also converted to azidation products

**Fig. 2 | Proposed mechanisms for the hydrolysis of Int-1.** Proposed mechanisms for hydrolysis.

(31–33) with moderate yields through the herein reported sequential process.

## Molecular editing of meta-azido pyridines

Inspired by nitrogen atom insertion from phenyl nitrene species,[31,32,35] we began our investigation using a range of meta-azido pyridines (Fig. 4). This N-insertion process offers different potential pathways, enabling access to structurally diverse products with markedly different characteristics. Pleasingly, we found that upon blue light irradiation in the presence of diethylamine as a nucleophile at room temperature, 2-phenyl meta-azido pyridine **1** was readily converted to the 1,3-diazepine **34** that was isolated in 54% yield. The structure of **34** was unambiguously assigned by X-ray crystal structure analysis. Pyridines carrying ortho-aryl substituents with electronic deficient halo and aldehyde groups at the 4-position provided the ring-enlarged 1,3-diazepines **35** (56%) and **36** (60%) in comparable yields. The N-insertion process is also feasible for the 4-phenyl pyridine-derived azide **7** to afford the diazepine **37** (58%). Of note, isomerization of the double bond occurred during ring-enlargement for this substrate. A similar outcome was observed for the para-fluorophenyl congener to deliver **38**. However, for the TMS-substituted para-phenyl derivative and a multi-substituted congener, significantly lower yields were achieved for the ring-enlargement isomerization cascade (**39, 40**). The lower yield is attributed to the poor stability of the product toward air-mediated oxidation. Notably, similar oxidative decomposition was also observed for other 1,3-diazepines, such as compounds **35** and **36**, which helps to explain the moderate yields obtained in these ring-enlargement processes.

The results became increasingly intriguing as we began probing the reactivity of disubstituted pyridyl azides. The azide derived from 3-methyl-2-phenylpyridine afforded the 1,3-diazepine **41** in good yield (65%). A comparable yield of 1,3-diazepine **42** (67%) was received with the more electron-rich methoxy-substituted azide **18**. In contrast, replacing the methyl group with a chloro substituent led predominantly to the formation of the 1,4-diazepine **43** (46%) via N-insertion between C(4) and C(5), along with 17% of the 1,3-diazepine **44**. 5-Azido-3-fluoro-2-phenylpyridine gave the 1,4-diazepine **45** as exclusive regioisomer in a moderate yield (31%) in combination with the azide reduction byproduct 5-fluoro-6-phenylpyridin-3-amine **46** (28%). The N-insertion in 5-azido-3-trifluoromethyl-2-phenylpyridine (**19**) also proceeded with excellent regioselectivity to give the 1,4-diazepine **47** in 51% yield, and the corresponding reduction product was not observed. Although the origin of the regio-divergent N-insertion as a function of the electronics of the oxazino–pyridine substituents is not yet understood, there is a trend: electron-deficient substituents at the 3-position of the pyridine favor the formation of 1,4-diazepines, whereas electron-rich ones preferentially yield 1,3-diazepines.

The photolysis of 2,5-disubstituted pyridyl azides exhibited markedly different reactivity compared to other substitution patterns. Upon irradiation under identical conditions, selective nitrogen insertion into an ortho C–H bond of the pendant aryl group occurred, rather than pyridine ring-enlargement, affording δ-carbolines **48–53** in good yields (55–77%). To note, the meta-chlorophenyl-substituted pyridine provided a mixture of the two possible regioisomers (see **53**, a:b = 1.2:1). Furthermore, we were pleased to observe the direct conversion of the pyridine ring to a 1,3-diazepine in both a typical pyridine ligand (**54**) and a more complex drug molecule (**55**), achieving 58% and 53% yield, respectively.

## Synthetic application and proposed mechanism

Larger-scale experiments and further synthetic transformations of pyridyl azides were conducted next, as demonstrated in Fig. 5. Unexpectedly, consistently higher yields for meta-azido pyridines were achieved when the reactions were scaled up (Fig. 5A). The improvement is deemed to the result of multiple factors involving hydrolysis and the following purification procedure. As shown in Table 1, a significant decrease in yield was attributed to the hydrolysis step and solvent also revealed great impact on the reaction efficiency. In the larger-scale reactions, we usually obtained foam-like adduct intermediates **Int-1** after drying under vacuum for few minutes, that was not observed in smaller scale runs. Therefore, we assume that residual solvent that is difficult to be fully removed in small-scale one-pot reactions might lead to diminished yields. The ring enlargement of compound **1** was also performed on a 2.5 mmol scale, affording the seven-membered diazepine **34** in 50% yield—comparable to that obtained in the small-scale experiment (Fig. 5, (a)). To further demonstrate the synthetic value of the meta-azido pyridines, additional established transformations involving the azide functionality were carried out, such as 1,3-cycloaddition (**56–58**) and reduction (**59**) (Fig. 5B, (b) (c) (d)).

A plausible mechanism exemplified for the functionalization of 2-phenyl pyridine is proposed in Fig. 5C. The oxazino pyridine **S1** is readily accessible from 2-phenyl pyridine through a known three-component reaction with dimethyl acetylene dicarboxylate and methyl pyruvate.[56] For the initial azidation step, we employed TEMPONa, a mild reducing reagent (SET) readily generated from commercially available TEMPO and sodium, for the generation of $N_3$ radicals from Zhdankin reagent **A**, and the persistent TEMPO radical is concomitantly formed during the process. The diene entity of the oxazino pyridine **S1** reacts with the electrophilic $N_3$ radical to generate the adduct carbon radical **60**, which is then selectively trapped by TEMPO to form **Int-1**. Subsequent hydrolysis and rearomatization under acidic condition eventually provide the meta-azido pyridine **1**.

Considering the mechanism of the ring-enlarging nitrogen insertion, a pyridinyl nitrene **61** is firstly generated from azide **1** upon blue light irradiation.[62] The reactive nitrene then inserts into the adjacent CC π bond on pyridine to yield a strained 2H-azirine **62**. For nitrene **61** insertion occurs regioselectively and the isomeric 2H-azirine **62'** does

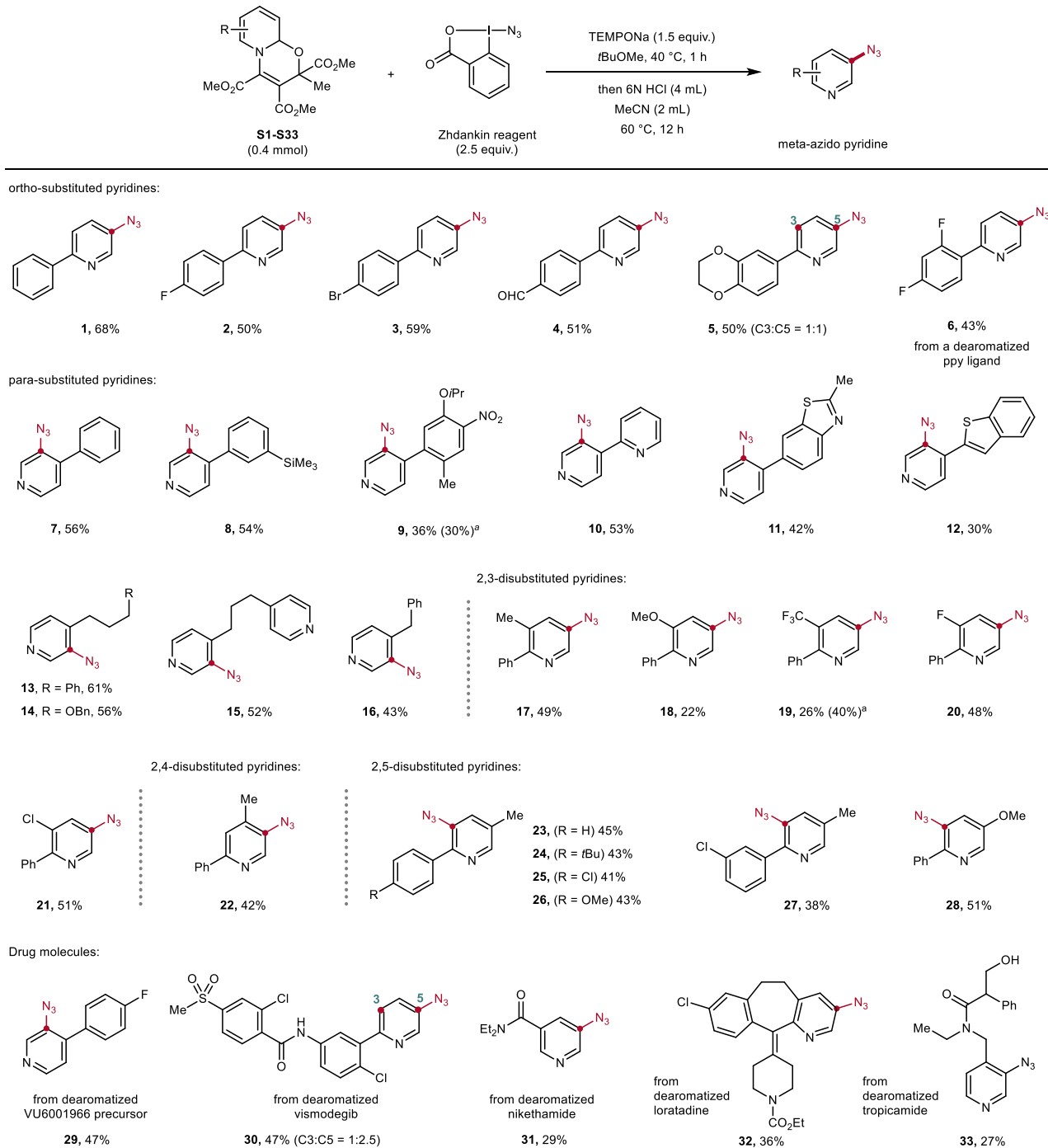

**Fig. 3 | Formal peripheral meta-azidation of pyridines.** Standard conditions: oxazino pyridine (0.4 mmol, 1.0 equiv), Zhdankin reagent (1.0 mmol, 2.5 equiv), TEMPONa (0.6 mmol, 1.5 equiv), *t*BuOMe (4.0 mL), 40 °C, 1 h; then 6 N HCl (4.0 mL), MeCN (2.0 mL), 60 °C, 12 h, isolated yield. *a* Starting material recovered.

not form. The 2H-azirine **62** then undergoes a thermal 6π electrocyclic ring opening to generate the electrophilic ketenimine **63**, which is finally trapped by the nucleophilic diethyl amine to deliver the isolated product 1,3-diazepine **34**.

Taken together, a sequential synthetic strategy has been developed to introduce nitrogen atoms into electron-deficient pyridines through peripheral meta-azidation and subsequent photo-mediated nitrene insertion. The first stage offers a reliable protocol for synthesizing meta-azido pyridines—compounds that are typically challenging to access—while demonstrating broad functional group tolerance and consistently good yields. Importantly, the obtained azides can be further diverted to diazepines and δ-carbolines through ring

expansion or ring fusion of pyridine-derived core structures via a nitrogen insertion process. The substitution pattern on the pyridine moiety directs the reactivity in the molecular editing step, leading to a range of N-heterocycles—including seven-membered 1,3-diazepines, 1,4-diazepines, and ring-fused δ-carbolines—all of which are biologically relevant scaffolds. In larger-scale transformations, reproducibility was demonstrated, along with the method's applicability to late-stage functionalization and structural editing of more complex substrates. This work overcomes long-standing synthetic challenges in accessing meta-azido pyridines and expands their utility in molecular editing applications. We are optimistic about the potential value of this approach in both pharmaceutical and synthetic applications.

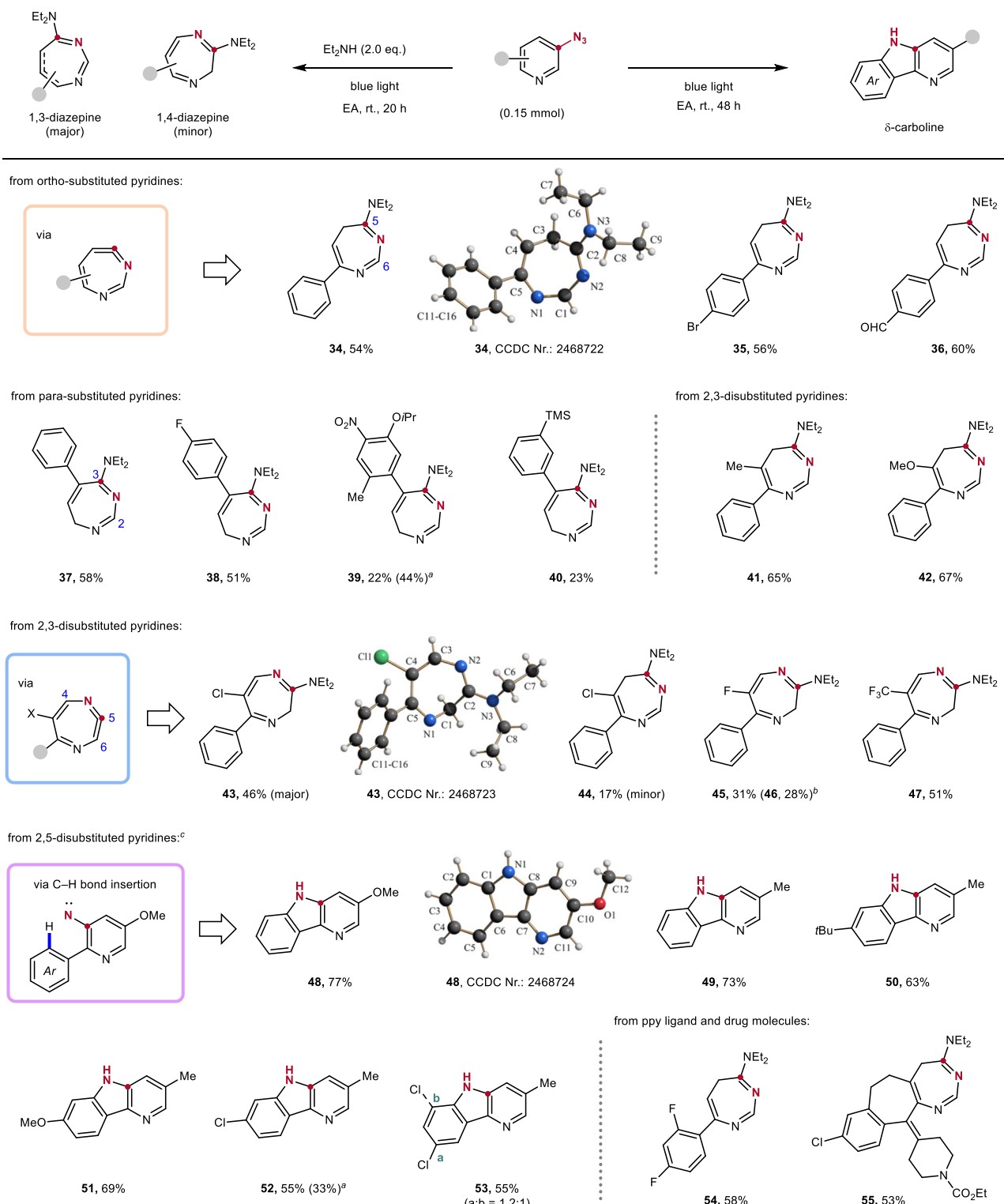

**Fig. 4 | Molecular editing of pyridines.** Standard conditions: pyridyl azide (0.15 mmol, 1.0 equiv), HNEt₂ (0.30 mmol, 2.0 equiv), EA (3.0 mL), blue light, 20 h, isolated yield. [a] Starting material recovered. [b] Isolated 5-fluoro-6-phenylpyridin-3-amine. [c] Without HNEt₂, 48 h.

## Methods

### General procedure for meta-C–H azidation

**Preparation of TEMPONa solution.** An oven-dried Schlenk tube was charged with freshly cleaned (with dry pentane) elemental sodium metal ($0.44 \times g$, 19 mmol, 1.4 equiv.), which was then melted to sodium mirror using a heating gun at 200 °C. The Schlenk tube was cooled to rt, and dry THF (16 mL), TEMPO ($2.12 \times g$, 13.6 mmol, 1 equiv.), and naphthalene (170 mg, 0.700 mmol, 0.1 equiv.) were added under argon atmosphere. The reaction mixture was stirred at room temperature until a blue-black color persisted (generally 1–2 h). TEMPONa solution can be stored under argon for several days in the fridge without any significant decomposition.

**Fig. 5 | Synthetic application and proposed mechanism. A** Larger-scale experiments. **B** Synthetic transformations of 5-azido-2-phenylpyridine (**1**). Conditions: (b) CuSO$_4$•5H$_2$O (10 mol%), sodium ascorbate (20 mol%), $t$BuOH/H$_2$O (0.1 M), rt, 18 h; (c) KOH (2.0 equiv.), EA, rt, 18 h; (d) H$_2$ (1 atm), Pd/C (10 mol%), MeOH, rt, 18 h. **C** Proposed mechanism.

**meta-C–H azidation of pyridines**. To an oven-dried 20 mL Schlenk tube, dearomatized pyridine (0.4 mmol, 1.0 equiv.) and 1-azido-1λ$^3$-benzo[d][1,2]iodaoxol-3(1H)-one **A** (1.0 mmol, 2.5 equiv.) were added. The vial was sealed with a septum and put under vacuum, followed by flushing with Ar gas 3 times. Tert-butyl methylether (4.0 mL, 0.1 M) was added to the mixture under Ar protection. The reaction mixture was heated with a water bath at 40 °C. Freshly prepared TEMPONa-solution (0.71 mL, 1.5 equiv., 0.85 M in THF) was added dropwise via syringe over the period of 30 min. Then, the reaction mixture was stirred at 40 °C for another 30 min.

**Caution.** For safety reasons, the reaction was carried out behind an anti-blast shield.

The reaction was filtered and washed with ethyl acetate (5 mL × 3). The filtrate was combined and the organic solvent was removed with a rotary evaporator under reduced pressure to give a colloidal residue. MeCN (2.0 mL) and 6 N HCl (4.0 mL) were then added to the residue and stirred at 60 °C for 12 h. Then, the reaction mixture was basified with saturated Na$_2$CO$_3$ aqueous solution and extracted with dichloromethane (50 mL, 3 times). The combined organic phase was dried over MgSO$_4$ and filtered. The solvent was removed under reduced pressure and the residue was purified by column chromatography on silica gel to give pure product.

**General procedure for ring enlargement**
To an oven-dried 5 mL Schlenk tube, pyridinyl azide (0.15 mmol, 1.0 equiv.) obtained through the azidation step was dissolved in 3.0 mL dry ethyl acetate, and diethyl amine (0.3 mmol, 2.0 equiv.) was added under Ar atmosphere. The vial was then sealed and put under vacuum, followed by flushing with Ar gas (3 times). The tube was placed in a photoreactor, stirred and irradiated for 20 h. The temperature was maintained below 30 °C using a fan. After completion of the reaction, as monitored by TLC, the solvent was removed under reduced pressure. The residue was purified by column chromatography (EtOAc/MeOH) on silica gel, to give the desired product. The concentrated product was dissolved in dichloromethane, then filtered with Nylon membrane filter (0.2 μm). Dichloromethane was removed under reduced pressure to give pure product.

**General procedure for C–H insertion**
To an oven-dried 5 mL Schlenk tube, pyridinyl azide (0.15 mmol, 1.0 equiv.) obtained through the azidation step was dissolved in 3.0 mL dry ethyl acetate under Ar atmosphere. The tube was then placed in a photoreactor, stirred, and irradiated for 48 h. The temperature was maintained below 30 °C using a fan. After the completion of the reaction, as monitored by TLC, the solvent was removed under reduced pressure. The residue was purified by column chromatography (pentane/EtOAc) on silica gel to give the pure product.

## Data availability
Details on the procedures and the corresponding datasets generated and/or analyzed in this study are provided in the paper and its Supplementary Information files, and from the corresponding author on request. The X-ray crystallographic coordinates for structures

reported in this study have been deposited at the Cambridge Crystallographic Data Centre (CCDC), under deposition numbers CCDC 2468722 (compound **34**), 2468723 (compound **43**), and 2468724 (compound **48**). These data can be obtained free of charge from The Cambridge Crystallographic Data Centre via www.ccdc.cam.ac.uk/data_request/cif.

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

## Acknowledgements
We thank Alexander von Humboldt foundation (fellowship to S.-M.G.) and the Deutsche Forschungsgemeinschaft for supporting this work. We thank Dr. K. Bergander, University of Münster, for conducting NMR experiments and Dr. M. Letzel and M. Haring, University of Münster, for MS analysis.

## Author contributions
S.-M.G. and A.S. conceived and designed the experiments. S.-M.G. and P.X. performed the experiments and analyzed the data. C.G.D. performed the X-ray analysis. S.-M.G. and A.S. wrote the manuscript.

## Funding

## Competing interests
The authors declare no competing interests.
