## [Transparent Peer Review file · Nature Communications]

Formal pyridine meta-azidation and its application for the synthesis of diazepines, ring-fused δ -carbolines and 1,2,3-triazolylpyridines

Corresponding Author: Professor Armido Studer

Version 0:

Reviewer comments:

Reviewer #1

(Remarks to the Author)

This study establishes a powerful two-stage strategy for pyridine diversification: unprecedented meta-selective azidation followed by substituent-dependent molecular editing of the resulting pyridyl azides. The observed divergence in azide transformation pathways—yielding distinct molecular architectures based on substitution patterns—creates a particularly interesting reaction manifold. This combined peripheral/molecular editing approach delivers exceptional diversity in nitrogen heterocycle synthesis from simple pyridine precursors, representing significant synthetic value. I recommend acceptance after the following issues have been addressed:

1. Cite important key recent related work on pyridine peripheral functionalization (Ye et al. *J. Am. Chem. Soc.*, 2011, 133, 6964; Fu et al. *Nat. Commun.* 2024, 15, 7420); pyridine-to-benzene molecular editing (Song et al. *CCS Chem.* 2025, 7, 392), pyridine-to-pyrrole skeletal editing (Fu et al. *Org. Lett.* 2020, 22, 6107);
2. The authors attribute the low yields of compounds 32 and 33 to air-mediated oxidation. To verify this explanation, it is suggested to perform the reaction under an inert atmosphere (e.g., N_2) and determine whether the yield improves.
3. To gain deeper mechanistic insights into the observed regioselectivity in the step of nitrene insertion of the 2, 3-disubstituted pyridyl azides, it would be valuable to systematically investigate electronic effects. This could involve preparing and testing azides derived from 3-substituted-2-phenylpyridines with electronically diverse substituents (e.g., F, CF_3 , OMe). Correlating the resulting 1,3- vs. 1,4-diazepine ratios with substituent parameters may help establish a rationale for the substituent-dependent divergence in nitrene insertion pathways.
4. Clarify "ortho-aryl group" insertion (Line 151) to specify "ortho C–H bonds of the pendant aryl group"
5. The compound number 32 in line 170 seems to be changed to 31.
6. Why must compounds 31-33 be analyzed by NMR at low temperatures, whereas compounds 34-37 do not require low-temperature conditions? Please explain the reason.
7. For Compound 31, the reported deuterated solvent in the NMR data is deuterated dichloromethane (CD_2Cl_2), while the NMR spectrum itself is labeled as using deuterated chloroform ($CDCl_3$). Please verify to ensure consistency.

Reviewer #2

(Remarks to the Author)

Studer and co-workers describe an azidation of oxazino-pyridines using the Zdhankin reagent and NaTEMPO. The resultant pyridine azide products are then transformed into heterocycles via photochemistry (1,3-diazepines) and carbazoles. Overall this manuscript shows a nice coherent picture of skeletal-editing of pyridines - the authors' lab has described numerous radical additions to these substrates in recent years, but the azidation is a new reaction and one with obvious application. The photo-transformations are well-known in general, but surprisingly unexplored for meta azidopyridines. Taken together the chemistry is relevant to current thinking and well exemplified in terms of interesting functional molecules.

It is quite unserious to present a paper on pyridine azidation and not include a single yield of pyridine azidation. All of the transformations are described from the oxazine derivative as if it is the starting material. This is very misleading for the products at the bottom of Figure 2 which are labelled 'from vismodegib, (47%)', 'from a ligand, (43%)' [what does a ligand mean?], as a reader would naturally assume these yields referred to FROM the starting pyridines when they do not. Please

correct this ambiguous representation and state the yields throughout Figure 2 from starting pyridines in line with the title of the paper.

Use of gram scale Zhdankin reagent is an explosion hazard and should be identified as such in the SI, in case chemists interested in using this method are unfamiliar with the reagent. Please cite Jerome Waser's work on the energetic qualities of this reagent.

Reviewer #3

(Remarks to the Author)

In this manuscript, Studer and co-workers report a synthetic pathway for the preparation of 3-azidopyridines from the corresponding pyridines. The authors employ an elegant one-pot, two-step dearomatization–rearomatization strategy. It should be noted, however, that this dearomatization intermediate has previously been leveraged multiple times to introduce various functional groups, including halogenation (Science 2022, 378, 779–785, DOI: 10.1126/science.ade6029), trifluoromethylation (J. Am. Chem. Soc. 2024, 146, 30758–30763), difluoromethylation (Nat. Commun. 15, 4121 (2024), <https://doi.org/10.1038/s41467-024-48383-1>), and nitration (J. Am. Chem. Soc. 2025, 147, 7485–7495). In the present work, the authors utilize the same intermediate to install an azido group.

Interestingly, many of the substrate examples involving the dearomatization pyridine intermediate overlap with those reported in the aforementioned studies. While this might suggest a certain level of redundancy, I admire Professor Studer's ability to repeatedly generate high-impact contributions by strategically modifying the introduced functional group with just one change. In my view, most of these studies could have been presented together as part of a broader meta-functionalization platform in a single article, rather than one functional group per paper.

In this study, the authors successfully access 3-azidopyridines and demonstrate the method in late-stage functionalization. While the application component is somewhat less novel—given that the skeletal editing of 3-azidopyridines was first reported by Wenstrup—the extension to a broader substrate scope and demonstration in late-stage settings is valuable. The scalability of the reaction to the millimole scale further underscores the practical utility of the method, as not all strategies are readily amenable to scale-up.

Overall, the main novelty lies in the installation of the azido group; therefore, I recommend acceptance after the following revisions and clarifications.

Specific Comments:

1. Scope and Functional Group Compatibility: Since the same intermediate has been employed multiple times with overlapping examples, it would strengthen the manuscript to include at least one example involving a drug molecule. Pyridine is a privileged scaffold in medicinal chemistry, and demonstrating functional group compatibility with motifs common in pharmaceuticals (e.g., free hydroxyl, carboxylic acid, or amine functionalities) would enhance the impact.

If the method is not compatible with these functionalities, a comment should be added regarding this limitation. Similarly, the tolerance toward alkenes and alkynes should be addressed; if possible, an example would be helpful. This would also better support the abstract statement: "Late-stage peripheral functionalization and structural editing of nitrogen-containing heterocycles have garnered increasing attention due to their potential for the preparation of diversifying drug-like libraries."

2. Regioisomer Formation in Example 28: In Example 28, both regioisomers are formed. Does the first step of dearomatization result in a regioisomeric mixture? If so, please clarify this point in the main text for the benefit of readers. Additionally, the procedure for S28 in the SI is not clearly provided. There appears to be no example listed after S24, unless it has been overlooked. This may be due to intermediates being reported in previous publications; however, the relevant NMR and characterization data should be included here or properly cited.

3. Figures:

i) In Figure 2, there is an extra red dot between 12 and 13; please remove it.

ii) In Figure 2, the structure of compound 30 (loratadine) is missing a pyridine ring nitrogen; please correct it.

iii) In Figure 4A, the structure of compound 30 (loratadine) is also missing a pyridine ring nitrogen atom; please correct it.

Version 1:

Reviewer comments:

Reviewer #1

(Remarks to the Author)

The authors have satisfactorily addressed all my concerns. The new experimental data and revised text significantly strengthen the manuscript. I recommend acceptance in its current form.

Reviewer #2

(Remarks to the Author)

The manuscript has been revised conscientiously and presents the meta-azidation protocol with more clarity, particularly with respect to the oxazino dearomatized intermediate. The inclusion of limitations further strengthens the usability of the protocol.

Reviewer #3

(Remarks to the Author)

I appreciate the author's thoughtful efforts in addressing my comments. All of my suggested revisions have been thoroughly incorporated.

Reviewer #1 (Remarks to the Author):

This study establishes a powerful two-stage strategy for pyridine diversification: unprecedented meta-selective azidation followed by substituent-dependent molecular editing of the resulting pyridyl azides. The observed divergence in azide transformation pathways—yielding distinct molecular architectures based on substitution patterns—creates a particularly interesting reaction manifold. This combined peripheral/molecular editing approach delivers exceptional diversity in nitrogen heterocycle synthesis from simple pyridine precursors, representing significant synthetic value. I recommend acceptance after the following issues have been addressed:

We thank this reviewer for the positive comments on our work. This reviewer mainly expressed some remarks on the interest of the methods, which we wish to address below.

1. Cite important key recent related work on pyridine peripheral functionalization (Ye et al. *J. Am. Chem. Soc.*, 2011, 133, 6964; Fu et al. *Nat. Commun.* 2024, 15, 7420); pyridine-to-benzene molecular editing (Song et al. *CCS Chem.* 2025, 7, 392), pyridine-to-pyrrole skeletal editing (Fu et al. *Org. Lett.* 2020, 22, 6107);

Answer: Thanks, these important works have been included as refs 24, 25, 26 and 27 into the revised version. The reference list was adapted accordingly.

2. The authors attribute the low yields of compounds 32 and 33 to air-mediated oxidation. To verify this explanation, it is suggested to perform the reaction under an inert atmosphere (e.g., N₂) and determine whether the yield improves.

Answer: Thanks for the suggestion. Indeed, all the reactions were performed under argon. We observed that some of the diazepines decomposed in the NMR tube after a few days, for example compound 35.

To further verify the effect of O₂, we also performed a control experiment by running first the reaction under standard condition. After completion, the mixture was flushed with O₂ for 5 min and then irradiation with blue light was continued for additional 4 hours. Product 34 fully decomposed in 4 hours. Two potential decomposition products (based on NMR) are shown below.

Control experiment:

3. To gain deeper mechanistic insights into the observed regioselectivity in the step of nitrene insertion of the 2,3-disubstituted pyridyl azides, it would be valuable to systematically investigate electronic effects. This could involve preparing and testing azides derived from 3-substituted-2-phenylpyridines with electronically diverse substituents (e.g., F, CF₃, OMe). Correlating the resulting 1,3- vs. 1,4-diazepine ratios with substituent parameters may help establish a rationale for the substituent-dependent divergence in nitrene insertion pathways.

Answer: Thanks for the constructive remark. We have synthesized the 5-azido-2-phenyl-3-(trifluoromethyl)pyridine (**19**). The N-insertion proceeded smoothly to deliver the product 1,4-diazepine **47** with 51% yield with complete regioselectivity. In addition, we also looked at the methoxy congener **18** that gave the 1,3-diazepine **42** as the only regioisomer. These results were added to the revised manuscript and SI.

The revised part reads as follows: "The azide derived from 3-methyl-2-phenylpyridine afforded the 1,3-diazepine **41** in good yield (65%). A comparable yield of 1,3-diazepine **42** (67%) was received with a more electron-rich methoxy-substituted azide (**18**). In contrast, replacing the methyl group with a chloro substituent led predominantly to the formation of the 1,4-diazepine **43** (46%) via N-insertion between C(4) and C(5), along with 17% of the 1,3-diazepine **44**. 5-Azido-3-fluoro-2-phenylpyridine gave the 1,4-diazepine **45** as exclusive regioisomer in a moderate yield (31%) in combination with the azide reduction byproduct 5-fluoro-6-phenylpyridin-3-amine **46** (28%). The N-insertion in 5-azido-3-trifluoromethyl-2-phenylpyridine (**19**) also proceeded with excellent regioselectivity to give the 1,4-diazepine **47** in 51% yield, and the corresponding reduction product was not observed. Although the origin of the regio-divergent N-insertion as a function of the electronics of the oxazino-pyridine substituents is not yet understood, there is a trend: electron-deficient substituents at the 3-position of the pyridine favor the formation of 1,4-diazepines, whereas electron-rich ones preferentially yield 1,3-diazepines."

4. Clarify "ortho-aryl group" insertion (Line 151) to specify "ortho C-H bonds of the pendant aryl group"

Answer: Thanks, corrected accordingly.

5. The compound number 32 in line 170 seems to be changed to 31.

Answer: Thanks, corrected in the text and Fig. 4.

6. Why must compounds 31-33 be analyzed by NMR at low temperatures, whereas compounds 34-37 do not require low-temperature conditions? Please explain the reason.

Answer: For products **31-33** (now **34-36**), we could not get well resolved NMR spectra at 25 °C, as line-broadening was observed for the methylene group (two different conformations that interconvert, no full coalescence at room temperature). In compounds **34-37** (now **37-40**) the protons of the methylene group are resolved at room temperature.

7. For Compound 31, the reported deuterated solvent in the NMR data is deuterated dichloromethane (CD_2Cl_2), while the NMR spectrum itself is labeled as using deuterated chloroform (CDCl_3). Please verify to ensure consistency.

Answer: Thanks, CD_2Cl_2 was used for compound **34** (former **31**), and it has been corrected in the SI.

Reviewer #2 (Remarks to the Author):

Studer and co-workers describe an azidation of oxazino-pyridines using the Zhdankin reagent and NaTEMPO. The resultant pyridine azide products are then transformed into heterocycles via photochemistry (1,3-azepines) and carbazoles.

Overall this manuscript shows a nice coherent picture of skeletal-editing of pyridines - the authors' lab has described numerous radical additions to these substrates in recent years, but the azidation is a new reaction and one with obvious application. The photo-transformations are well-known in general, but

surprisingly unexplored for meta azidopyridines. Taken together the chemistry is relevant to current thinking and well exemplified in terms of interesting functional molecules.

We thank this reviewer for the positive feedback on our research.

It is quite unserious to present a paper on pyridine azidation and not include a single yield of pyridine azidation. All of the transformations are described from the oxazine derivative as if it is the starting material. This is very misleading for the products at the bottom of Figure 2 which are labelled 'from vismodegib, (47%)', 'from a ligand, (43%)' [what does a ligand mean?], as a reader would naturally assume these yields referred to FROM the starting pyridines when they do not. Please correct this ambiguous representation and state the yields throughout Figure 2 from starting pyridines in line with the title of the paper.

Answer: Thanks for the comment. We agree and changed the title to “Formal pyridine meta-azidation and its application for the synthesis of diazepines, ring-fused δ -carboline and 1,2,3-triazolopyridines”. We reworded accordingly at several places in the manuscript to further clarify that we run the azidation on oxazino pyridines. Ligand means that this particular pyridine is often used as a ligand in catalysis. The yields in Figure 2 refer to yields starting from oxazino pyridines that also appear as the substrates in the reaction scheme of Figure 2. To further clarify that we use the oxazino pyridines as the starting materials we stated in the text that “All oxazino pyridines used in this study were prepared according to literature^{51,53,54,57,62} in 88% average yield (see SI).” As discussed in the paper, the rearomatization step of the azido-oxazino pyridines turned out to be challenging and solvent-dependent. Therefore, we were not able to develop a good one pot protocol that also includes the initial dearomatization step.

Use of gram scale Zhdankin reagent is an explosion hazard and should be identified as such in the SI, in case chemists interested in using this method are unfamiliar with the reagent. Please cite Jerome Waser's work on the energetic qualities of this reagent.

Answer: Thanks for the important comment. We have added the following sentence in the SI, **GP2** part. “**Caution: For safety reasons, the reaction was carried out behind an anti-blast shield.**” In addition, the work of Prof. Jerome Waser (*J. Org. Chem.* **2018**, *83*, 12334–12356.) was also cited at the end of the sentence.

Reviewer #3 (Remarks to the Author):

In this manuscript, Studer and co-workers report a synthetic pathway for the preparation of 3-azidopyridines from the corresponding pyridines. The authors employ an elegant one-pot, two-step dearomatization–rearomatization strategy. It should be noted, however, that this dearomatization intermediate has previously been leveraged multiple times to introduce various functional groups, including halogenation (*Science* **2022**, *378*, 779–785, DOI: 10.1126/science.ade6029), trifluoromethylation (*J. Am. Chem. Soc.* **2024**, *146*, 30758–30763), difluoromethylation (*Nat. Commun.* **15**, 4121 (2024), <https://doi.org/10.1038/s41467-024-48383-1>), and nitration (*J. Am. Chem. Soc.* **2025**, *147*, 7485–7495). In the present work, the authors utilize the same intermediate to install an azido group.

Interestingly, many of the substrate examples involving the dearomatization pyridine intermediate overlap with those reported in the aforementioned studies. While this might suggest a certain level of

redundancy, I admire Professor Studer's ability to repeatedly generate high-impact contributions by strategically modifying the introduced functional group with just one change. In my view, most of these studies could have been presented together as part of a broader meta-functionalization platform in a single article, rather than one functional group per paper.

In this study, the authors successfully access 3-azidopyridines and demonstrate the method in late-stage functionalization. While the application component is somewhat less novel—given that the skeletal editing of 3-azidopyridines was first reported by Wentrup—the extension to a broader substrate scope and demonstration in late-stage settings is valuable. The scalability of the reaction to the millimole scale further underscores the practical utility of the method, as not all strategies are readily amenable to scale-up.

Overall, the main novelty lies in the installation of the azido group; therefore, I recommend acceptance after the following revisions and clarifications.

We thank this reviewer for the thorough evaluation of our research and constructive remarks on our manuscript, which led to an improved version.

Specific Comments:

1. Scope and Functional Group Compatibility: Since the same intermediate has been employed multiple times with overlapping examples, it would strengthen the manuscript to include at least one example involving a drug molecule. Pyridine is a privileged scaffold in medicinal chemistry, and demonstrating functional group compatibility with motifs common in pharmaceuticals (e.g., free hydroxyl, carboxylic acid, or amine functionalities) would enhance the impact.

If the method is not compatible with these functionalities, a comment should be added regarding this limitation. Similarly, the tolerance toward alkenes and alkynes should be addressed; if possible, an example would be helpful. This would also better support the abstract statement: "Late-stage peripheral functionalization and structural editing of nitrogen-containing heterocycles have garnered increasing attention due to their potential for the preparation of diversifying drug-like libraries."

Answer: Thanks for the constructive remark. We synthesized additional new substrates and performed the azidation reactions with **GP2**. For a drug containing a free alcohol, we are able to access the *meta*-azidated tropicamide (**33**) in 27% yield. See revised manuscript and SI.

Unfortunately, substrates containing an alkene or alkyne functionality failed to give the desired azides due to the multi-reactive sites resulting in a complex mixture as judged by TLC. Substrates containing a free carboxylic acid and a free amine are not compatible with the TEMPO_{Na} reaction system. We have presented the failed cases also in SI (see page S21).

Failed cases:

2. Regioisomer Formation in Example 28: In Example 28, both regioisomers are formed. Does the first step of dearomatization result in a regioisomeric mixture? If so, please clarify this point in the main text for the benefit of readers. Additionally, the procedure for S28 in the SI is not clearly provided. There appears to be no example listed after S24, unless it has been overlooked. This may be due to intermediates being reported in previous publications; however, the relevant NMR and characterization data should be included here or properly cited.

Answer: Thanks for the comments. The dearomatization of ortho-aryl substituted pyridines (such as pyridine **S30**, former **S28**) can only provide a single regioisomer in the dearomatization step. The procedure for **S30** was added to the SI. NMR data and characterization data of the previously reported oxazino pyridines were added. Further, we also provide the original references for these compounds in the revised SI.

3. Figures:

i) In Figure 2, there is an extra red dot between 12 and 13; please remove it.

Answer: Thanks, it is removed.

ii) In Figure 2, the structure of compound 30 (loratadine) is missing a pyridine ring nitrogen; please correct it.

Answer: Thanks, it is corrected.

iii) In Figure 4A, the structure of compound 30 (loratadine) is also missing a pyridine ring nitrogen atom; please correct it.

Answer: Thanks, it is corrected.